# Serological Survey of Mosquito-Borne Arboviruses in Wild Birds from Important Migratory Hotspots in Romania

**DOI:** 10.3390/pathogens11111270

**Published:** 2022-10-31

**Authors:** Mircea Coroian, Cornelia Silaghi, Birke Andrea Tews, Emanuel Ștefan Baltag, Mihai Marinov, Vasile Alexe, Zsuzsa Kalmár, Horváth Cintia, Mihaela Sorina Lupșe, Andrei Daniel Mihalca

**Affiliations:** 1Department of Parasitology and Parasitic Diseases, University of Agricultural Sciences and Veterinary Medicine of Cluj-Napoca, 400372 Cluj-Napoca, Romania; 2Department of Infectious Diseases, “Iuliu Hatieganu” University of Medicine and Pharmacy, 400000 Cluj-Napoca, Romania; 3Institute of Infectology, Friedrich-Loeffler-Institut, Federal Research Institute for Animal Health, D-17493 Greifswald, Germany; 4Marine Biological Stationof Agigea, University “Alexandru Ioan Cuza” of Iași, 907018 Iași, Romania; 5Danube Delta National Institute for Research and Development, 820112 Tulcea, Romania; 6Department of Microbiology, Immunology and Epidemiology, University of Agricultural Sciences and Veterinary Medicine of Cluj-Napoca, 400372 Cluj-Napoca, Romania; 7Clinical Hospital of Infectious Diseases, 400337 Cluj-Napoca, Romania; 8ELKH-ÁTE Climate Change: New Blood-Sucking Parasites and Vector-Borne Pathogens Research Group, 1078 Budapest, Hungary

**Keywords:** West Nile virus, Sindbis virus, Usutu virus, arboviruses, vector-borne diseases

## Abstract

In the context of climate change, globalization, and enhanced human traveling, arboviruses continue to represent a threat to public health. West Nile and Usutu viruses are mosquito-borne viruses belonging to the *Flaviviridae* family (*Flavivirus* genus) and members of the Japanese encephalitis virus serocomplex. Included in the *Togaviridae* family (*Alphavirus* genus), the Sindbis virus is also vectored by mosquitoes. In the present study, we aimed to analyze the presence of antibodies concerning the abovementioned viruses in migratory and resident birds in the South-Eastern region of Romania, as avian hosts represent the main reservoir for human infection. Blood samples were collected from wild birds between May 2018 and October 2019 in nine locations from three counties. All the samples were serologically tested by ELISA and a serum neutralization test. Overall, a seroprevalence of 8.72% was registered for the West Nile virus, 2.71% for the Usutu virus, and 0% for the Sindbis virus. To our best knowledge, this is the first large-scale comprehensive study to assess the West Nile virus seropositivity in wild birds and the first serological confirmation of the Usutu virus in wild birds in Romania. Moreover, this is the only follow-up study reviewing the current seroprevalence of the Sindbis virus in Romania since 1975.

## 1. Introduction

In the context of climate change, globalization, and enhanced human traveling, arboviruses continue to pose a threat to public health [1]. Birds are important reservoirs for arboviruses such as the West Nile (WNV) and Usutu (USUV) viruses. These are mosquito-borne viruses belonging to the *Flaviviridae* family (*Flavivirus* genus) and members of the Japanese encephalitis virus serocomplex. The Sindbis virus (SINV), also vectored by mosquitoes, is included in the *Togaviridae* family (*Alphavirus* genus) [2].

Since its first report in Romania in 1996, WNV circulation has been recorded yearly, with significant outbreaks among humans [3,4]. Serological evidence was also recorded during the following years in domestic and wild birds [5,6]. Although WNV causes disease and mortality every year, in Romania, it remains passively monitored solely in humans, birds, and horses, with acute neurological cases being tested.

The first introduction of USUV in Europe was described in Austria in 2001, associated with the high mortality of European blackbirds (*Turdus merula*) and great gray owls (*Strix nebulosa*), followed by a retrospective study in Italy on bird tissue samples [7,8]. To date, the virus circulation has been recorded in birds from several European countries, such as Hungary, Italy, and Germany [9]. In Romania, USUV antibodies were recently documented for the first time in a domestic dog, but their presence in humans, birds, or other susceptible hosts has not yet been demonstrated [10]. In addition, the co-circulation of WNV and USUV was also reported in Europe in 30 species of birds belonging to 11 orders [11].

SINV was reported for the first time in Europe in 1965 when antibodies were detected in humans from Italy and Finland and in birds in the Volga Delta [12,13]. The only report from Romania dates back to 1975 when a low seroprevalence was being registered in humans [14].

The medical importance of arboviruses lies in their zoonotic potential. In humans, WNV usually evolves asymptomatically or with flu-like symptoms. In 1% of the infections, humans develop the West Nile neuroinvasive disease, which is associated with meningoencephalitis and, in some cases, fatalities [15,16]. USUV was also found to be responsible for severe neuro-invasive infections in humans [17]. SINV has mostly been associated with disease in humans, and so far, the infections were reported in the northern part of Europe (Sweden, Finland, and Russia) and South Africa [18]. The disease is associated with fever, arthritis, and a skin rash [19]. However, USUV and SINV are not considered in the surveillance strategy of Romania.

These viruses have common features, sharing aspects of ecology and epidemiology. Their life cycle is accomplished by ornithophilic mosquitoes that act as vectors and vertebrate hosts. The birds are susceptible to reservoir hosts due to the high and prolonged viremia they develop [18].

Avian migration is considered to be of great medical importance because of the ability of birds to carry, spread, and transmit a wide range of pathogens, such as viruses, bacteria, fungi, and parasites [20]. Migratory birds are considered responsible for the introduction and long-distance spread of the arboviruses, while resident birds maintain the viral amplification and local circulation [21]. Located on the western route of the Eurasian-East African flyway, which connects the Black Sea with the African continent, Romania is an important stopover for migratory birds [22]. Due to the habitat characteristics, which include wetlands and estuaries, previous studies have mentioned the Danube Delta as being a hotspot for WNV outbreaks in Romania. This habitat is not only favorable for birds but also for the reproduction of mosquitoes [21]. However, no studies on the circulation of mosquito-borne avian-associated arboviruses in such areas or areas with multi-annual human cases have been published to date.

The aim of our study was to assess the seroprevalence of WNV, USUV, and SINV in blood samples from wild birds and in endemic areas for mosquito-borne viruses from Romania.

## 2. Materials and Methods

### 2.1. Sample Collection

The study was conducted in the southeastern part of Romania between May 2018 and October 2019, as this is one of the major avian migration hotspots in Romania and also the area where most human cases are reported [23]. The sampling protocol included nine locations in three counties: Constanța, Tulcea counties, and Bucharest.

Blood samples (max 1% of body weight) were collected from wild birds, which were captured by using standard mist nets. The following data were recorded for each bird: the date and location of capture, species, age, and gender. Depending on the migratory behavior of the wild birds, they were divided into long-distance and short-distance migrants. Blood samples were collected from the jugular vein and centrifuged at 8000 rpm for 10 min. The serum was collected and stored at −80 °C until further analysis.

### 2.2. Serological Analysis

#### 2.2.1. ELISA

All serum samples were screened at the Department of Parasitology and Parasitic Diseases of the University of Agricultural Sciences and Veterinary Medicine of Cluj-Napoca, using the ELISA method for WNV. We used the commercial kit INGEZIM West Nile COMPAC (Eurofins Technologies, Madrid, Spain), targeting specific antibodies against protein E. The method was performed according to the manufacturer’s instructions.

#### 2.2.2. Viruses and Cells

Vero cells (L0015 Collection of Cell Lines in Veterinary Medicine (CCLV), Friedrich- Loeffler-Institut, Greifswald—Insel Riems, Germany) were routinely cultivated in a minimal essential medium (MEM: MEM Hank’s Salts/MEM Earles’ Salts, New York, NY, USA), with 10% fetal calf serum (FCS) at 37 °C with 5% CO_2_. The viruses used were WNV lineage II Germany/2019/T167-20 (LR743455.1, MN921233), Usutu Europe 3 (HE599647), and Sindbis Edsbyn.

#### 2.2.3. Serum Neutralization Test (SNT)

Depending on the volume of sera available from the birds with positive or equivocal IgG ELISA index values for WNV, serum samples were further analyzed by SNT for WNV, USUV, and SINV. All sera were heat-inactivated before the analyses (30 min at 56 °C) at the Friedrich-Loeffler-Institut, Germany.

SNT was performed in microtiter plates as described in the OIE terrestrial manual. Briefly, two-fold dilution series starting with a 1:5 dilution was made on all the serum samples. Serum dilutions were incubated with 100 Tcid_50_ of the respective virus for 1 h at 37 °C. After incubation, 1 × 10^4^ Vero cells were added to each well, and samples were incubated for 7 days. Cells were fixed and stained with crystal violet to help with cytopathic effect detection. If the volume allowed, serum samples were tested in duplicates with a WNV lineage II, USUV lineage Europe 3, and SINV. For the low-quantity samples, a chosen subset of the viruses was tested. Neutralizing titres were calculated as the geometric mean of the duplicates.

Sera were considered positive if they showed neutralizing titres equal to 1:10 or higher. Between the WNV and USUV sera, they were considered clearly positive for one virus but not the other if the neutralization titre was four-fold higher for that virus compared to the other.

### 2.3. Statistical Analysis

The statistical analysis was performed using EpiInfoTM 2000 software (version 7.2.0.1., Atlanta, Georgia). The prevalence of WNV and USUV was estimated from the ratio of positive to the total number of samples, with the exact binomial confidence intervals of 95%. We excluded from this analysis the bird species with less than 10 samples.

A Generalized Linear Model (GLM) with a log link function approach was used to determine the directly relative risk of infection of the studied birds with WNV and USUV. For the analysis, we used the positive cases and the total number of samples collected in each location. GLM analysis was used to find which species was the most suitable to be infected among the assessed species with WNV or with USUV.

### 2.4. Ethical Statement

Live, apparently healthy birds were captured using mist nets and released immediately after sampling. These birds were captured according to the OM 1380/13.07.2019 by specially trained personnel, with ringing permits (No. 740622) from the Romanian Ornithological Centre, Academy of Agriculture and Forestry.

## 3. Results

### 3.1. Study Group

The study group consisted of 1304 serum samples collected from wild birds in three counties from Romania, as follows: two locations in Constanța (Agigea Bird Observatory, Grindul Lupilor), six locations in the Tulcea Counties (Batag, Bididia, Enisala, Murighiol, Sălcioara, Tulcea) and one location in Bucharest (Figure 1, Table 1).

The birds were sampled between May 2018 (*n* = 521; 40.0%; 95% CI: 37.3–42.6) and October 2019 (*n* = 783; 60.1%, 95% CI: 57.4–62.7), and belonged to 46 species from 33 genera, represented by 22 families and six orders (Table 2).

Overall, the majority of the samples were collected from house sparrows *(Passer domesticus*, 37.5%), followed by the Eurasian reed warbler (*Acrocephalus scirpaceus*, 14.7%). Out of the 1304 birds included in the study, 691 (53.0%; 95% CI: 50.3–55.7) were adults, 439 (33.7%; 95% CI: 31.2–36.3) were juveniles, and in the case of the remaining 174 individuals (13.4%; 95% CI: 11.6–15.3), the age could not be assessed. Additionally, 249 (19.1%; 95% CI: 17.1–21.3) birds were females, 220 (16.9%; 95% CI: 14.9–19.0) were males, and for 835 (64.0%; 95% CI: 61.4–66.6) birds the gender could not be assessed.

### 3.2. ELISA

From the total of the 1304 samples screened for WNV by ELISA, 11.5% (150; 95% CI: 9.9–13.4) tested positive, and 82.8% returned negative results (1080; 95% CI: 80.7–84.8). Additionally, 5.7% of the sera were considered doubtful (74; 95% CI: 4.5–7.1) (Table 3).

### 3.3. Serum-Neutralization Test (SNT)

#### 3.3.1. WNV SNT

Due to the low volume of some of the samples, out of 224 (17.2%; 95% CI: 15.2–19.3) positive or doubtful IgG ELISA samples, 182 (14.0%; 95% CI: 12.2–15.9) were tested by SNT. A total of 11 samples that were positive in ELISA could not be confirmed through SNT. By contrast, nine samples that were doubtful in ELISA showed neutralization in SNT. Overall, a seroprevalence of 8.7% (110/1262; 95% CI: 7.3–10.4) was registered by SNT. Seroconversion was present in 19 species of wild birds. In the analysis samples that tested positive for WNV and USUV in SNT and that did not show a four-fold difference between the neutralization titre, allowing a clear decision for one virus and the classification of the other signal as a cross-reaction, were actually counted as positive for both viruses. A total of 55 tested samples for both viruses could be identified as having a four-fold higher neutralization to WNV, and four could be identified as having a four-fold higher reaction to USUV. Another 32 samples that tested positive for WNV could not be tested against USUV due to sample scarcity. Prevalence by species was analyzed using GLM. According to this analysis, the blackbird (*Turdus merula*) was the most suitable to be infected with WNV (*p* = 0.03) (Table 4).

Regarding the gender of the birds, a seroprevalence of 12.6% (29/231; 95% CI: 8.6–17.5) was recorded in females and 9.8% (21/215; 95% CI: 6.2–14.5) in the males. In the samples for which the gender could not be assessed, a seroprevalence of 7.4% (60/816; 95% CI: 5.8–9.4) was registered. In age groups, we recorded a seroprevalence of 7.4% (32/431; 95% CI: 5.3–10.3) in juveniles and 7.5% (50/666; 95% CI: 5.7–9.8) in adults. For the birds where the age could not be assessed, we recorded a positivity rate of 17.0% (28/165; 95% CI: 11.6–23.6).

Additionally, out of nine localities, eight were positive for WNV (Table 5). Sera that could be identified as being truly WNV positive (with a four-fold higher SN titre against WNV than USUV) came from all regions. The positivity rates recorded in 2018 were 7.6% (38/501; 95% CI: 5.6%–10.2), and 9.5% (72/761; 95% CI: 7.6–11.8%) in 2019.

Depending on the migratory behavior, the seroprevalence rates of WNV and USUV by SNT are described in Table 6.

#### 3.3.2. USUV SNT

Of the positive or doubtful IgG ELISA samples for WNV, a total of 136 (10.4%; 95% CI: 8.9–12.2) were tested for USUV by SNT. We recorded a seroprevalence of 2.7% (33/1216; 95% CI: 1.9–3.8). Antibodies were detected in nine wild bird species in five localities. Overall, nine species and five localities were positive for both WNV and USUV (Table 4 and Table 5). Of the four uniquely USUV-positive samples, two were from Grindul Lupilor, and two were from Sălcioara.

No significant differences in seroprevalence were identified in 2018 (2.1%, 10/483; 95% CI: 1.1–3.8) or 2019 (3.1%, 23/733; 95% CI: 2.1–4.7). Additionally, within the age groups, the seroprevalence was 2.7% (17/634; 95% CI: 1.7–4.3) in adults and 2.6% (11/422; 95% CI: 1.5–4.6) in juveniles. Birds for which the age could not be assessed recorded a seroprevalence of 3.1% (5/160; 95% CI: 1.0–7.1). For gender, the seroprevalence was 2.7% (6/221; 95% CI: 1.0–5.8) in females and 3.9% (8/207; 95% CI: 1.7–7.5) in males. Of the samples for which the gender could not be assessed a seroprevalence of 2.4% (19/788; 95% CI: 1.6–3.7) was registered.

#### 3.3.3. Double Reacting Sera

A total of 2.4% of the samples showed neutralization titres against WNV and USUV without meeting the criteria for a four-fold higher titre against one virus or the other. Some showed titres comparable to those found in sera that reacted mainly to one virus (e.g., 1/900 or 1/640), whereas others showed lower titres. These samples came from nine wild bird species and five localities (Table 7). No statistically significant results were registered between the samples from 2018 (2.1%, 10/483; 95% CI: 1.1–3.8) and 2019 (2.6%, 19/733; 95% CI: 1.7–4.0).

Additionally, co-reacting sera were registered for the age groups as follows: 2.1% in adults (13/634; 95% CI: 1.2–3.5), 2.6% in juveniles (11/422; 95% CI: 1.5–4.61), and 3.13% in the category where age was undetermined (5/160; 95% CI: 1.02–7.14). As for gender, the coinfections rates were: 1.8% in females (4/221; 95% CI: 0.5–4.6), 3.86% in males (8/207; 95% CI: 1.68–7.47), and 2.16% in the uncategorized samples (17/788; 95% CI: 1.35–3.43).

According to GLM, the highest risk of infection was with WNV (*p* < 0.01), followed by USUV (*p* = 0.6).

#### 3.3.4. SINV SNT

Of the total samples, 5.36% (70/1304) were tested by SNT for SINV. All of the assessed samples returned negative results.

## 4. Discussion

WNV, USUV, and SINV are mosquito-borne viruses with continuous circulation in Europe. Given the heterogeneity of surveillance methods used in Europe and the cross-reactions that may influence the diagnosis, these viruses remain a threat to public health [24,25]. As previously stated, the plaque-reduction neutralization test (PRNT) is recognized as the gold standard method for the serological diagnostics of flaviviruses [26]. However, Gennaro et al. concluded that serum neutralization tests could successfully replace the PRNT [27]. Even in serum neutralization tests that relate flaviviruses from the same sero-complex, they cannot be easily distinguished. Thus, infection with one virus leads to cross-reactive antibodies against other viruses in the same sero-complex. In the present study, we aimed to analyze the presence of antibodies for WNV, USUV, and SINV in migratory and resident birds from the southeastern region of Romania. The study was conducted in an area overlapping the main migration routes of wild birds. We also considered Bucharest, the capital city and an endemic area with the highest predicted probability for WNV infections in humans in Romania [23].

The SNT for WNV in this study was comparable with those from other studies on wild birds conducted in Romania. Previous studies conducted on birds in Romania have shown seroprevalence rates of 3.5% and 8.8% [5,28]. Another study from Romania reported a seroprevalence of 32.1% by ELISA, but no further confirmation tests were performed [6]. In addition to the previous data, the present study included a much larger number of samples belonging to 46 species of wild birds.

This is the first report of seropositivity to USUV in wild birds from Romania. This is supported by the fact that the sera reacted much more strongly against USUV than WNV. Similar to WNV, USUV has the same life cycle, using birds as amplifying hosts for the introduction and spread of the virus [29]. Recently, USUV antibodies were confirmed in a dog from Romania [10]. Several sera (2.4%) in this study could not be clearly distinguished as WNV- or USUV-positive as they reacted to both viruses with similar titers in neutralization assays. These could either be due to an infection with yet another virus in the same sero-complex or due to infections with both viruses. Co-circulation is likely as birds whose sera react only to USUV, or much better to USUV, were sampled in locations that also yielded sera reacting only to WNV. This emphasizes their co-circulation, which was already mentioned in previous studies conducted in Europe [11,30]. Several studies have previously attempted to find the correlation between coinfections with the aforementioned viruses and the immune response [31]. Given that both viruses can determine/induce lesions incompatible with life on their own, it would prove quite a challenge to correlate their symptomatology. Moreover, Zhou et al., and Wang et al., discussed the appearance of a superinfection exclusion phenomenon which would determine the inhibition of one of the viruses (in this case, WNV would inhibit the productive replication of USUV within the cell) [32,33]. However, additional studies are needed to confirm these theories.

Previous studies suggested a possible sentinel role of the common blackbird in outbreaks of USUV infections, as the virus introduction in Europe was associated with increased mortality in this bird species in all the countries where outbreaks were recorded [7,30,34,35]. A possible explanation could be the weaker immune system of this species [36]. The results we obtained in this study suggest the same in the case of WNV, as previously observed in a study conducted in Germany [37]. Given the recent clinical cases in humans, we suggest testing cases of viral encephalitis for USUV and SINV as well [17]. Moreover, the confirmation of WNV in blood donors in several European countries should be taken into consideration by the public institutions responsible for the blood test methodology in transfusions [4,38].

The presence of SINV in wildlife has already been reported in Europe [39]. Moreover, the co-circulation of SINV with WNV and USUV was also registered in birds influenced by similar aspects of the life cycle, suggesting similar epidemiological mechanisms [2]. Since 1975 only one study concerning SINV in Romania was conducted, revealing a prevalence of 0.6% in humans [14]. The complete absence of a serological response in our study could be related to the relatively small sample size.

To our best knowledge, this is the first large-scale comprehensive study to assess the WNV seropositivity and the first serological confirmation of USUV in wild birds in Romania. Moreover, this is the only study assessing the current seroprevalence of SINV in Romania since 1975.

## 5. Conclusions

Our study brings new information on the eco-epidemiology and co-circulation of WNV and USUV in Romania, emphasizing the important role of wild birds in their life cycle. Regarding SINV, despite the low seroprevalence, further studies are required in order to establish its epidemiological status in Romania.

## Figures and Tables

**Figure 1 pathogens-11-01270-f001:**
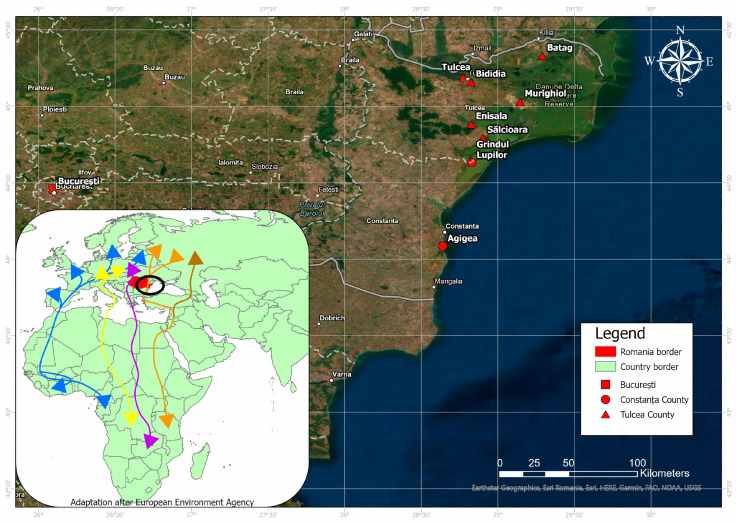
Sampling localities and the main migration routes (adaptation after European Environment Agency) for the wild birds. The colored arrows represent the most important migration routes of wild birds.

**Table 1 pathogens-11-01270-t001:** Sampling locations.

County	Location	Sampling Prevalence % (+/*n*; 95% CI)
Tulcea	Batag	14.6 (80/550; 11.8–17.7)
	Bididia	4.0 (22/550; 2.7–5.6)
	Enisala	7.6 (42/550; 5.7–10.2)
	Murighiol	6.9 (38/550; 5.1–9.3)
	Sălcioara	44.2 (243/550; 40.1–48.4)
	Tulcea	22.7 (125/550; 19.4–26.4)
	Total	42.2 (550/1304; 46.7–52.1)
Constanța	Agigea	58.2 (375/644; 54.4–62.0)
	Grindul Lupilor	41.2 (269/644; 38.0–45.6)
	Total	49.4 (644/1304; 46.7–52.1)
Bucharest	București	8.4 (110/110; 7.1–10.1)
	Total	8.4 (110/1304; 7.1–10.1)

+/*n*: number of samples from that location/total number of samples.

**Table 2 pathogens-11-01270-t002:** The collected bird species according to their taxonomic classification.

Order	Family	Genus	Species	Prevalence % (+/*n*; 95% CI)
Bucerotiformes	Upupidae	*Upupa*	*Upupa epops*	1.4 (18/1304; 0.9–2.2)
Columbiformes	Columbidae	*Streptopelia*	*Streptopelia decaocto*	0.3 (4/1304; 0.1–0.8)
Coraciiformes	Meropidae	*Merops*	*Merops apiaster*	0.2 (2/1304; 0.1–0.6)
Galliformes	Phasianidae	*Phasianus*	*Phasianus colchicus*	0.1 (1/1304; 0.0–0.4)
Passeriformes	Acrocephalidae	*Acrocephalus*	*Acrocephalus agricola*	1.9 (25/1304; 1.3–2.8)
			*Acrocephalus arudinaceus*	0.5 (7/1304; 0.3–1.1)
			*Acrocephalus schoenobaenus*	0.1 (1/1304; 0.0–0.4)
			*Acrocephalus scirpaceus*	14.7 (191/1304; 12.8–16.7)
		*Hippolais*	*Hippolais icterina*	0.3 (4/1304; 0.1–0.8)
	Corvidae	*Corvus*	*Corvus monedula*	0.2 (2/1304; 0.0–0.6)
		*Pica*	*Pica pica*	0.1 (1/1304; 0.0–0.4)
		*Garrulus*	*Garrulus glandarius*	0.1 (1/1304; 0.0–0.4)
	Emberizidae	*Emberiza*	*Emberiza schoeniclus*	1.1 (14/1304; 0.6–1.8)
	Fringillidae	*Carduelis*	*Carduelis chloris*	3.4 (44/1304; 2.5–4.5)
		*Coccothraustes*	*Coccothraustes coccothraustes*	0.1 (1/1304; 0.0–0.4)
	Hirundinidae	*Hirundo*	*Hirundo rustica*	0.2 (3/1304; 0.08–0.67)
	Laniidae	*Lanius*	*Lanius collurio*	3.2 (42/1304; 2.39–4.32)
	Locustellidae	*Locustella*	*Locustella luscinoides*	0.2 (3/1304; 0.1–0.7)
			*Locustella fluviatilis*	0.1 (1/1304; 0.0–0.4)
	Motacillidae	*Anthus*	*Anthus trivialis*	0.2 (2/1304; 0.1–0.6)
		*Motacilla*	*Motacilla alba*	0.1 (1/1304; 0.0–0.4)
	Muscicapidae	*Erithacus*	*Erithacus rubecula*	1.2 (15/1304; 0.7–1.9)
		*Ficedula*	*Ficedula albicollis*	0.1 (1/1304; 0.01–0.43)
		*Luscinia*	*Luscinia megarinchos*	0.5 (6/1304; 0.2–1.0)
			*Luscinia luscinia*	0.4 (5/1304; 0.2–0.9)
		*Muscicapa*	*Muscicapa striata*	1.5 (19/1304; 0.9–2.3)
		*Phoenicurus*	*Phoenicurus phoenicurus*	2.2 (29/1304; 1.6–3.2)
		*Saxicola*	*Saxicola rubetra*	0.2 (2/1304; 0.1–0.6)
	Oriolidae	*Oriolus*	*Oriolus oriolus*	0.1 (1/1304; 0.0–0.4)
	Paridae	*Parus*	*Parus major*	0.2 (2/1304; 0.1–0.6)
	Passeridae	*Passer*	*Passer domesticus*	37.5 (489/1304; 33.9–41.5)
			*Passer montanus*	8.6 (112/1304; 6.9–10.8)
	Phylloscopidae	*Phylloscopus*	*Phylloscopus trochillus*	0.1 (1/1304; 0.0–0.4)
	Prunellidae	*Prunella*	*Prunella modularis*	0.2 (3/1304; 0.1–0.7)
	Sturnidae	*Sturnus*	*Sturnus vulgaris*	0.7 (9/1304; 0.4–1.3)
	Sylviidae	*Sylvia*	*Sylvia atricapilla*	5.4 (70/1304; 4.3–6.7)
			*Sylvia borin*	2.1 (27/1304; 1.4–3.0)
			*Sylvia communis*	0.7 (9/1304; 0.4–1.3)
			*Sylvia curruca*	0.8 (11/1304; 0.5–1.5)
			*Sylvia nissoria*	0.2 (2/1304; 0.1–0.6)
	Turdidae	*Turdus*	*Turdus iliacus*	0.1 (1/1304; 0.01–0.43)
			*Turdus merula*	3.6 (47/1304; 2.7–4.8)
			*Turdus philomelos*	5.3 (69/1304; 4.2–6.7)
Piciformes	Picidae	*Dendrocopos*	*Dendrocopos minor*	0.2 (2/1304; 0.1–0.6)
			*Dendrocopos syriacus*	0.2 (3/1304; 0.1–0.7)
		*Jynx*	*Jynx torquilla*	0.2 (2/1304; 0.1–0.6)

+/*n*: number of samples of that species/total number of samples.

**Table 3 pathogens-11-01270-t003:** ELISA positive and doubtful results for WNV antibodies.

Species	Prevalence % (+/*n*; 95% CI)	
	Positive	Doubtful
*Acrocephalus agricola*	12 (3/25; 2.6–31.2)	16.0 (4/25, 4.5–36.1)
*Acrocephalus arudinaceus*	0 (0/7)	0 (0/7)
*Acrocephalus schoenobaenus*	0 (0/1)	0 (0/1)
*Acrocephalus scirpaceus*	7.9 (15/191; 4.5–12.6)	12.6 (24/191; 8.2–18.1)
*Anthus trivialis*	0 (0/2)	0 (0/2)
*Carduelis chloris*	6.8 (3/44; 1.4–18.7)	4.5 (2/44; 0.6–15.5)
*Coccothraustes coccothraustes*	0 (0/1)	0 (0/1)
*Corvus monedula*	50.0 (1/2; 1.3–98.7)	0 (0/2)
*Dendrocopos minor*	0 (0/2)	0 (0/2)
*Dendrocopos syriacus*	0 (0/3)	0 (0/3)
*Emberiza schoeniclus*	0 (0/14)	0 (0/14)
*Erithacus rubecula*	6.7 (1/15; 0.2–32.0)	6.7 (1/15; 0.2–32.0)
*Ficedula albicollis*	0 (0/1)	0 (0/1)
*Garrulus glandarius*	0 (0/1)	0 (0/1)
*Hippolais icterina*	3.3 (1/33; 0.8–90.6)	0 (0/3)
*Hirundo rustica*	0 (0/3)	33.3 (1/3; 0.8–90.6)
*Jynx torquilla*	0 (0/2)	0 (0/2)
*Lanius collurio*	11.9 (5/42; 4.0–25.6)	0 (0/42)
*Locustella fluviatilis*	0 (0/1)	0 (0/1)
*Locustella luscinoides*	0 (0/3)	0 (0/3)
*Luscinia luscinia*	0 (0/5)	20.0 (1/5; 0.5–71.6)
*Luscinia megarinchos*	0 (0/6)	16.7 (1/6; 0.4–64.1)
*Merops apiaster*	50.0 (1/2; 1.3–98.7))	0/2 (0)
*Motacilla alba*	100.0 (1/1; 2.5–100.0)	0/1 (0)
*Muscicapa striata*	15.8 (3/19; 3.4–39.6)	0/19 (0)
*Oriolus oriolus*	100.0 (1/1; 2.5–100.0)	0/1 (0)
*Parus major*	50.0 (1/2; 1.3–98.7)	0/2 (0)
*Passer domesticus*	13.5 (66/489; 10.8–16.8)	4.3 (21/489; 2.8–6.5)
*Passer montanus*	4.5 (5/112; 1.5–10.1)	4.5 (5/112; 1.5–10.1)
*Phasianus colchicus*	0 (0/1)	100.0 (1/1; 2.5–100.0)
*Phoenicurus phoenicurus*	0 (0/29)	10.3 (3/29; 2.2–27.4)
*Phylloscopus trochillus*	0 (0/1)	0 (0/1)
*Pica pica*	0 (0/1)	0 (0/1)
*Prunella modularis*	0 (0/3)	0 (0/3)
*Saxicola rubetra*	0 (0/2)	0 (0/2)
*Streptopelia decaocto*	0 (0/4)	0 (0/4)
*Sturnus vulgaris*	0 (0/9)	0 (0/9)
*Sylvia atricapilla*	0 (0/70)	4.3 (3/70; 0.9–12.0)
*Sylvia borin*	0 (0/27)	0 (0/27)
*Sylvia communis*	22.2 (2/9; 2.8–60.0)	11.1 (1/9; 0.3–48.3)
*Sylvia curruca*	9.1 (1/11; 0.2–41.3)	0 (0/11)
*Sylvia nissoria*	50.0 (1/2; 1.3–98.7)	0 (0/2)
*Turdus iliacus*	0 (0/1)	0 (0/1)
*Turdus merula*	48.9 (23/47; 34.1–63.9)	2.2 (1/47; 0.1–11.3)
*Turdus philomelos*	14.5 (10/69; 7.2–25.0)	2.9 (2/69; 0.4–10.1)
*Upupa epops*	33.3 (6/18; 13.3–59.0)	16.7 (3/18; 3.6–41.4)
Total	11.5 (150/1304; 10.0–13.4)	5.7 (74/1304; 4.5–7.1)

+/*n*: number of ELISA positive samples/total number of samples.

**Table 4 pathogens-11-01270-t004:** WNV and USUV SNT seropositivity rates among the assessed bird species.

Species	Prevalence % (+/*n*; 95% CI)	
	WNV SNT	USUV SNT
*Acrocephalus agricola*	12.0 (3/25; 2.6–31.2)	4.6 (1/22; 0.1–22.8)
*Acrocephalus scirpaceus*	6.5 (12/185; 3.4–11.1)	2.9 (5/174; 0.9–6.6)
*Carduelis chloris*	4.7 (2/43; 0.6–15.8)	0 (0/43)
*Corvus monedula*	50 (1/2; 1.26–98.74)	0 (0/2)
*Erithacus rubecula*	6.7 (1/15; 0.2–32.0)	0 (0/15)
*Hirundo rustica*	33.3 (1/3; 0.8–90.6)	0 (0/2)
*Lanius collurio*	7.3 (3/41; 1.5–19.9)	4.9 (2/41; 0.0–16.3)
*Merops apiaster*	50.0 (1/2; 1.3–98.7)	50.0 (1/2; 1.3–98.4)
*Motacilla alba*	100.0 (1/1; 2.5–100.0)	nt
*Muscicapa striata*	11.1 (2/18; 1.4–34.7)	0 (0/17)
*Parus major*	50.0 (1/2; 1.3–98.7)	0 (0/2)
*Passer domesticus*	9.6 (45/469; 7.3–12.6)	3.8 (17/450; 2.4–6.0)
*Passer montanus*	2.7 (3/110; 0.6–7.8)	0.9 (1/109; 0.0–5.0)
*Phoenicurus phoenicurus*	6.9 (2/29; 0.9–22.8)	0 (0/28)
*Sylvia communis*	25.0 (2/8; 3.2–65.1)	0 (0/7)
*Sylvia nissoria*	50.0 (1/2; 1.3–98.7)	0 (0/2)
*Turdus merula*	40.5 (17/42; 25.6–56.7)	7.5 (3/40; 1.6–20.4)
*Turdus philomelos*	11.8 (8/68; 5.2–21.9)	3.1 (2/65; 0.4–10.7)
*Upupa epops*	25.0 (4/16; 7.3–52.4)	6.3 (1/16; 0.2–30.2)
Total	8.7 (110/1262; 7.3–10.4)	2.7 (33/1216; 1.9–3.8)

+/*n*: number of SNT positive samples/total number of samples, nt—not tested for USUV NT.

**Table 5 pathogens-11-01270-t005:** WNV and USUV SNT seropositivity rates for the analyzed localities.

Locality (County)	Prevalence % (+/*n*; 95% CI)	
	WNV SNT	USUV SNT
Agigea (CT)	11.0 (40/364; 8.2–14.6)	2.3 (8/355; 1.2–4.4)
Batag (TL)	1.3 (1/77; 0.0–7.0)	0 (0/76; 95.3–100.0)
Bididia (TL)	4.6 (1/22; 0.1–22.8)	0 (0/22; 84.6–100.0)
București (B)	6.6 (7/106; 2.7–13.1)	0 (0/102; 96.5–100.0)
Enisala (TL)	0 (0/41; 91.4–100.0)	0 (0/41; 91.4–100.0)
Grindul Lupilor (CT)	6.1 (16/261; 3.5–9.8)	2.9 (7/246; 1.2–5.8)
Murighiol (TL)	22.9 (8/35; 10.4–40.1)	9.4 (3/32; 2.0–25.0)
Sălcioara (TL)	8.5 (20/236; 5.3–12.8)	5.2 (12/232; 2.7–8.9)
Tulcea (TL)	14.2 (17/120; 8.5–21.7)	2.7 (3/110; 0.6–7.8)
Total	8.7 (110/1262; 7.3–10.4)	2.7 (33/1216; 1.9–3.8)

+/*n*: number of positive samples/total number of samples.

**Table 6 pathogens-11-01270-t006:** SNT prevalence rates of WNV and USUV depending on the migratory behavior of the wild birds assessed in Romania.

Wild Bird Migration Behavior	Prevalence % (+/*n*; 95% CI)	
	WNV	USUV
Long-distance migrants	7.5 (31/415; 5.3–10.4)	2.4 (10/415; 1.3–4.4)
Short-distance migrants	12.7 (27/213; 8.85–17.81)	2.3 (5/213; 1.4–5.4)
Local movements	7.7 (52/676; 5.9–9.9)	2.7 (18/676, 1.7–4.2)
Total	8.7 (110/1262; 7.3–10.4)	2.7 (33/1216; 1.9–3.8)

+/*n*: number of positive samples/total number of samples.

**Table 7 pathogens-11-01270-t007:** WNV and USUV co-reaction rates in SNT for the wild bird species assessed.

Wild Bird Species	Prevalence % (+/*n*; 95% CI)
*Acrocephalus agricola*	4.6 (1/22; 0.1–22.8)
*Acrocephalus scirpaceus*	1.7 (3/174; 0.4–5.0)
*Lanius collurio*	4.9 (2/41; 0.6–16.5)
*Merops apiaster*	50.0 (1/2; 1.3–98.7)
*Passer domesticus*	3.3 (15/450; 2.0–5.4)
*Passer montanus*	0.9 (1/109; 0.0–5.0)
*Turdus merula*	7.5 (3/40; 1.6–20.4)
*Turdus philomelos*	3.1 (2/65; 0.4–10.7)
*Upupa epops*	6.3 (1/16; 0.2–30.2)
Total	2.4 (29/1216; 1.7–3.4)

+/*n*: number of positive samples/total number of samples.

## Data Availability

All data generated or analysed during this study are included in this published article. Other datasets used and/or analysed can be made available by the corresponding author on reasonable request.

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
