# Peer review of "Serological Survey of Mosquito-Borne Arboviruses in Wild Birds from Important Migratory Hotspots in Romania"

_pathogens, 2022, doi:10.3390/pathogens11111270_

Round 1

Reviewer 1 Report

The article « Serological survey of mosquito-borne arboviruses in wild birds from important migratory hotspots in Romania » detects WNV and USUV circulation in wild birds in Romania and the absence of antibodies against the Sindbis virus. While the authors have done a great job collecting samples from birds in different areas in Romania, I was wondering why they only opted for serology, while PCR testing and sequencing would have provided complementary data on the real prevalence of these viruses and the circulating strains and lineages, among other advantages. The manuscript is hard to follow as logical links are sometimes missing. Added to a general restructuring, the manuscript would also benefit from extensive corrections to delete unnecessary repetitions in the abstract, introduction, discussion, and conclusion parts. Please consider these changes :

Line 23: Add viruses

Lines 24 and 25: Please put the virus Family and genus in italics. Same remark in the introduction. Latin names should be in italic also.

Abstract: Please indicate why this survey is important both to animal and human health.

Lines 45-46: It is unlikely that human infections preceded that of domestic and wild birds. Please adapt accordingly.

Lines 50-51: The second part of the sentence sounds incomplete. Could you explain the retrospective study results? Were blackbirds the only species found positive?

Line 54 and 71: Do you mean susceptible hosts?

Line 55: Why did the authors bring up WNV here? Otherwise, USUV alone can infect at least 93 bird species from 35 families (Benzarti et al., Journal of General Virology 2019;100:119–132).

If co-circulation is important, please explain the context. Has there been any surveillance report concerning WNV or USUV in mosquitoes?

Line 61: please start a new paragraph to deal with the medical importance of the 3 arboviruses.

Line 67: Please move the following sentence here, lines 56-57 « USUV was also found responsible for severe neuro-invasive infections in humans »

Lines 75-78: This is very broad and irrelevant to the article, please delete it.

Line 95: Please indicate the blood volume sampled from each bird.

Line 1O5: Please indicate the antigen targeted by this ELISA.

What is the rationale behind using these tests? Why not include universal or specific PCR tests to detect acute cases that could be overlooked using serology, especially during the hot season when blood samples were collected?

Line 113: Please provide data about the Sindbis virus strain used.

General remark: please re-check the abbreviations in this manuscript that have not been enounced (MEM, FCS, TCID50, etc…)

Lines 38-39: Could the authors explain the logical link between the first and the second part of the sentence? Same in lines 231-233.

Line 174: Was any of the ELISA-positive samples negative using VNT? There is no part in the discussion confronting both test results?

Line 245: Please add a reference.

Lines 251: Not only in Austria but wherever USUV outbreaks were reported: France, Germany, Italy, Belgium…

Lines 251-252: Could the authors better explain this?

Lines 257-259: Why evoking WNV and USUV here?

Line 260: Low prevalence or complete absence of serological response?

Author Response

Reviewer 1 (R1)

R1: The article « Serological survey of mosquito-borne arboviruses in wild birds from important migratory hotspots in Romania » detects WNV and USUV circulation in wild birds in Romania and the absence of antibodies against the Sindbis virus. While the authors have done a great job collecting samples from birds in different areas in Romania, I was wondering why they only opted for serology, while PCR testing and sequencing would have provided complementary data on the real prevalence of these viruses and the circulating strains and lineages, among other advantages.

The manuscript is hard to follow as logical links are sometimes missing. Added to a general restructuring, the manuscript would also benefit from extensive corrections to delete unnecessary repetitions in the abstract, introduction, discussion, and conclusion parts. Please consider these changes:

Line 23: Add viruses

Author response (AR): done.

R1: Lines 24 and 25: Please put the virus Family and genus in italics. Same remark in the introduction. Latin names should be in italic also.

AR: done.

R1: Abstract: Please indicate why this survey is important both to animal and human health.

AR: done.

R1: Lines 45-46: It is unlikely that human infections preceded that of domestic and wild birds. Please adapt accordingly.

AR: We do not state that human infection preceded the infection in birds, it is only the chronology of studies in Romania.

R1: Lines 50-51: The second part of the sentence sounds incomplete. Could you explain the retrospective study results? Were blackbirds the only species found positive?

AR: rephrased.

R1: Line 54 and 71: Do you mean susceptible hosts?

AR: adjusted.

R1: Line 55: Why did the authors bring up WNV here? Otherwise, USUV alone can infect at least 93 bird species from 35 families (Benzarti et al., Journal of General Virology 2019;100:119–132). If co-circulation is important, please explain the context. Has there been any surveillance report concerning WNV or USUV in mosquitoes?

AR: rephrased.

R1: Line 61: please start a new paragraph to deal with the medical importance of the 3 arboviruses.

AR: done

R1: Line 67: Please move the following sentence here, lines 56-57 « USUV was also found responsible for severe neuro-invasive infections in humans »

AR: done

R1: Lines 75-78: This is very broad and irrelevant to the article, please delete it.

AR: Part of the paragraph content was deleted; the remaining was made more specific and merged with the following one.

R1: Line 95: Please indicate the blood volume sampled from each bird.

AR: done

R1: Line 1O5: Please indicate the antigen targeted by this ELISA.

AR: done

R1: What is the rationale behind using these tests? Why not include universal or specific PCR tests to detect acute cases that could be overlooked using serology, especially during the hot season when blood samples were collected?

AR: Sampling sites included remote areas, where ensuring preservation conditions to prevent RNA denaturation were lacking (-80ᵒ). Moreover, the purpose of this study was surveillance, and we considered that the serological method is more appropriate, taking into account the short-term viremia that the birds develop.

R1: Line 113: Please provide data about the Sindbis virus strain used.

AR: done.

R1: General remark: please re-check the abbreviations in this manuscript that have not been enounced (MEM, FCS, TCID50, etc…)

AR: done.

R1: Lines 38-39: Could the authors explain the logical link between the first and the second part of the sentence? Same in lines 231-233.

AR: It is unclear what the reviewer asks here.

R1: Line 174: Was any of the ELISA-positive samples negative using VNT? There is no part in the discussion confronting both test results?

AR: Done. We have mention this in the results section.

R1: Line 245: Please add a reference.

AR: Added: Vilibic-Cavlek T, Petrovic T, Savic V, Barbic L, Tabain I, Stevanovic V, Klobucar A, Mrzljak A, Ilic M, Bogdanic M, Benvin I, Santini M, Capak K, Monaco F, Listes E, Savini G. Epidemiology of Usutu Virus: The European Scenario. Pathogens. 2020 Aug 26;9(9):699. doi: 10.3390/pathogens9090699. PMID: 32858963; PMCID: PMC7560012.

R1: Lines 251: Not only in Austria but wherever USUV outbreaks were reported: France, Germany, Italy, Belgium…

AR: Adjusted.

R1: Lines 251-252: Could the authors better explain this?

AR: rephrased.

R1: Lines 257-259: Why evoking WNV and USUV here?

AR: rephrased.

R1: Line 260: Low prevalence or complete absence of serological response?

AR: corrected.

Review 2 (R2)

R1: Title: Serological survey of mosquito-borne arboviruses in wild birds from important migratory hotspots in Romania

In this manuscript the authors investigated the presence of antibodies against arboviruses in migratory and resident birds in the south-eastern region of Romania. This manuscript describes an important and growing public health problem in Europe. It is hard to accurately predict the spread of arboviruses worldwide. Therefore, such large-scale studies are important to improve predictions and protection against these viruses.

With the globalization and climate change, it is expected faster spread of mosquito-borne arboviruses around the world. Therefore, the serological survey of West Nine Virus (WNV) and Usutu Virus (USUV), but also the development of diagnostic applications for arboviruses will play an even more important role in the health of animals and humans.

I recommend this paper for publication with minor editing changes suggested below:

Major points

A critical point in serological studies is cross-reactivity between different viruses. Thus, the authors should describe in more details how they can exclude cross-reactivities and false positive results. This central question should be discussed in more detail in this paper.

Author response (AR): Added: In order to exclude possible cross-reactions, we used serum neutralization assay. As previously stated, plaque-reduction neutralization test (PRNT) is recognized as the gold standard method for the serological diagnostic of flaviviruses (Sambri et al., 2013). However, Gennaro concluded that serum neutralization test can replace successfully the PRNT (Gennaro et al., 2014). Lines: 242-246. We are aware that cross reactive antibodies are raised by different viruses in the same serocomplex. We added several discussions of this fact in several places in the manuscript to reflect this fact.

R2: The data about coinfections of WNV and USUV virus is interesting: Could pre-existing USUV infection influence the immune response and antibody production against WNV?

AR: We consider this is a general mechanism, as any comorbidity could influence the immune response of the guest.

R2: Could the viremia and transmission of WNV be reduced in USUV infected animals?

AR: I’ve addressed this question in the manuscript. Line: 291-299.

R2: This is a critical point in studies involving USUV and WNV and should be better discussed because impacts in the significance of this manuscript.

AR: This is indeed a very interesting question, thank you for raising it. I agree with your point of view but additional studies are needed to confirm these theories.

Reviewer 2 Report

Title: Serological survey of mosquito-borne arboviruses in wild birds from important migratory hotspots in Romania

In this manuscript the authors investigated the presence of antibodies against arboviruses in migratory and resident birds in the south-eastern region of Romania. This manuscript describes an important and growing public health problem in Europe. It is hard to accurately predict the spread of arboviruses worldwide. Therefore, such large-scale studies are important to improve predictions and protection against these viruses.

With the globalization and climate change, it is expected faster spread of mosquito-borne arboviruses around the world. Therefore, the serological survey of West Nine Virus (WNV) and Usutu Virus (USUV), but also the development of diagnostic applications for arboviruses will play an even more important role in the health of animals and humans.

I recommend this paper for publication with minor editing changes suggested below:

Major points

A critical point in serological studies is cross-reactivity between different viruses. Thus, the authors should describe in more details how they can exclude cross-reactivities and false positive results. This central question should be discussed in more detail in this paper.

The data about coinfections of WNV and USUV virus is interesting: Could pre-existing USUV infection influence the immune response and antibody production against WNV? Could the viremia and transmission of WNV be reduced in USUV infected animals? This is a critical point in studies involving USUV and WNV and should be better discussed because impacts in the significance of this manuscript.

Minor points:

Line 31: “…this is the first large-scale comprehensive study to assess the West Nile virus seropositivity in wild birds”. The authors should explain in more detail in the discussion, the difference between previous studies conducted in Romania (lines 82 and 239) and this large-scale study here.

Line 52: “…recorded in several European countries such as Hungary, Italy, and Germany”. It would be also interesting the comparison between the numbers founded in Romania and other countries.

Line 68: “…our country”. Please, reword to Romania.

Line 82: “…previous studies have mentioned the Danube Delta as being a hotspot for WNV outbreaks in the country”. See above, line 31.

Lines 93-94: “…three counties: Constanta, and Tulcea counties, and Bucharest.”.

Please, reword this sentence to “three counties: Constanta, Tulcea counties, and Bucharest”. Why did the authors choose these counties? Are the data in these counties comparable to other regions in Romania? Do the authors expect similar results in other counties in Romania? Please, discuss this in more detail.

Line 122: “…1x104 Vero cells”. 1x10^4 Vero cells?

Line 159: “…Passer domesticus, 37,5%”. In table 2 Passer domesticus is = 38%. 38% or 37,5%?

Line 177: “…the blackbird (Turdus merula) is the most suitable to be infected with WNV”. Could the authors explain why the blackbird is higher infected with WNV?

Lines 181-182: “…a seroprevalence of 12.6% was recorded in females and 9,8% in males.” Is this difference between males and females expected? Or is there another explanation? For example, behavior?

Lines 189: “…Positivity rate recorded in 2018 was 7,6% and 9,5% in 2019.” Is this difference expected? Or the positivity rate is increasing? How do the authors explain this difference?

Line 212: “Coinfections”. See above, major points.

Line 222: “…1.8% in females, 3,86% in males.” See above, line 181.

Line 239: “…our country”. See above, line 68.

Line 239: “…Previous studies conducted on birds in Romania”. See above, lines 31 and 82.

Line 254-256: “…the confirmation of WNV in blood donors in several European countries should be taken into consideration by the public institutions responsible for the blood test methodology in transfusions.”

A very important point for the public health. The authors could also discuss the importance of developing serological tests to avoid false positive results, for a better protection of animals and humans.

Author Response

Reviewer 2 (R2)

R1: Title: Serological survey of mosquito-borne arboviruses in wild birds from important migratory hotspots in Romania

In this manuscript the authors investigated the presence of antibodies against arboviruses in migratory and resident birds in the south-eastern region of Romania. This manuscript describes an important and growing public health problem in Europe. It is hard to accurately predict the spread of arboviruses worldwide. Therefore, such large-scale studies are important to improve predictions and protection against these viruses.

With the globalization and climate change, it is expected faster spread of mosquito-borne arboviruses around the world. Therefore, the serological survey of West Nine Virus (WNV) and Usutu Virus (USUV), but also the development of diagnostic applications for arboviruses will play an even more important role in the health of animals and humans.

I recommend this paper for publication with minor editing changes suggested below:

Major points

A critical point in serological studies is cross-reactivity between different viruses. Thus, the authors should describe in more details how they can exclude cross-reactivities and false positive results. This central question should be discussed in more detail in this paper.

Author response (AR): Added: In order to exclude possible cross-reactions, we used serum neutralization assay. As previously stated, plaque-reduction neutralization test (PRNT) is recognized as the gold standard method for the serological diagnostic of flaviviruses (Sambri et al., 2013). However, Gennaro concluded that serum neutralization test can replace successfully the PRNT (Gennaro et al., 2014). Lines: 242-246. We are aware that cross reactive antibodies are raised by different viruses in the same serocomplex. We added several discussions of this fact in several places in the manuscript to reflect this fact.

R2: The data about coinfections of WNV and USUV virus is interesting: Could pre-existing USUV infection influence the immune response and antibody production against WNV?

AR: We consider this is a general mechanism, as any comorbidity could influence the immune response of the guest.

R2: Could the viremia and transmission of WNV be reduced in USUV infected animals?

AR: I’ve addressed this question in the manuscript. Line: 291-299.

R2: This is a critical point in studies involving USUV and WNV and should be better discussed because impacts in the significance of this manuscript.

AR: This is indeed a very interesting question, thank you for raising it. I agree with your point of view but additional studies are needed to confirm these theories.

R2: Minor points:

Line 31: “…this is the first large-scale comprehensive study to assess the West Nile virus seropositivity in wild birds”. The authors should explain in more detail in the discussion, the difference between previous studies conducted in Romania (lines 82 and 239) and this large-scale study here.

AR: done.

R2: Line 52: “…recorded in several European countries such as Hungary, Italy, and Germany”. It would be also interesting the comparison between the numbers founded in Romania and other countries.

AR: In Romania there was not study in birds prior to our study.

R2: Line 68: “…our country”. Please, reword to Romania.

AR: Done.

R2: Line 82: “…previous studies have mentioned the Danube Delta as being a hotspot for WNV outbreaks in the country”. See above, line 31.

AR: Done.

R2: Lines 93-94: “…three counties: Constanta, and Tulcea counties, and Bucharest.” Please, reword this sentence to “three counties: Constanta, Tulcea counties, and Bucharest”.

AR: Done.

R2: Why did the authors choose these counties? Are the data in these counties comparable to other regions in Romania? Do the authors expect similar results in other counties in Romania? Please, discuss this in more detail.

AR: Done.

R2: Line 122: “…1x104 Vero cells”. 1x10^4 Vero cells?

AR: Done.

R2: Line 159: “…Passer domesticus, 37,5%”. In table 2 Passer domesticus is = 38%. 38% or 37,5%?

AR: Corrected in the table.

R2: Line 177: “…the blackbird (Turdus merula) is the most suitable to be infected with WNV”. Could the authors explain why the blackbird is higher infected with WNV?

AR: Added.

R2: Lines 181-182: “…a seroprevalence of 12.6% was recorded in females and 9,8% in males.” Is this difference between males and females expected? Or is there another explanation? For example, behavior?

AR: As this difference is not statistically significant, as shown by our analysis, this was not discussed.

R2: Lines 189: “…Positivity rate recorded in 2018 was 7,6% and 9,5% in 2019.” Is this difference expected? Or the positivity rate is increasing? How do the authors explain this difference?

AR: As this difference is not statistically significant, as shown by our analysis, this was not discussed.

R2: Line 212: “Coinfections”. See above, major points.

AR: See above.

R2: Line 222: “…1.8% in females, 3,86% in males.” See above, line 181.

AR: As this difference is not statistically significant, as shown by our analysis, this was not discussed.

R2: Line 239: “…our country”. See above, line 68.

AR: Done.

R2: Line 239: “…Previous studies conducted on birds in Romania”. See above, lines 31 and 82.

AR: Done, as suggested above.

R2: Line 254-256: “…the confirmation of WNV in blood donors in several European countries should be taken into consideration by the public institutions responsible for the blood test methodology in transfusions.” A very important point for the public health. The authors could also discuss the importance of developing serological tests to avoid false positive results, for a better protection of animals and humans.

AR: Indeed, this is a very important point but totally unrelated to our study.